# Rational Zika vaccine design via the modulation of antigen membrane anchors in chimpanzee adenoviral vectors

César López-Camacho [1], Peter Abbink [2], Rafael A. Larocca [2], Wanwisa Dejnirattisai[3], Michael Boyd[2], Alex Badamchi-Zadeh[2], Zoë R. Wallace[4], Jennifer Doig[5], Ricardo Sanchez Velazquez [5], Roberto Dias Lins Neto [6], Danilo F. Coelho [6], Young Chan Kim [1], Claire L. Donald [5], Ania Owsianka[5], Giuditta De Lorenzo[5], Alain Kohl[5], Sarah C. Gilbert [7], Lucy Dorrell[4], Juthathip Mongkolsapaya[3,8], Arvind H. Patel [5], Gavin R. Screaton[9], Dan H. Barouch[2], Adrian V.S. Hill [7] & Arturo Reyes-Sandoval [1]

Zika virus (ZIKV) emerged on a global scale and no licensed vaccine ensures long-lasting anti-ZIKV immunity. Here we report the design and comparative evaluation of four replication-deficient chimpanzee adenoviral (ChAdOx1) ZIKV vaccine candidates comprising the addition or deletion of precursor membrane (prM) and envelope, with or without its transmembrane domain (TM). A single, non-adjuvanted vaccination of ChAdOx1 ZIKV vaccines elicits suitable levels of protective responses in mice challenged with ZIKV. ChAdOx1 prME ΔTM encoding prM and envelope without TM provides 100% protection, as well as long-lasting anti-envelope immune responses and no evidence of in vitro antibody-dependent enhancement to dengue virus. Deletion of prM and addition of TM reduces protective efficacy and yields lower anti-envelope responses. Our finding that immunity against ZIKV can be enhanced by modulating antigen membrane anchoring highlights important parameters in the design of viral vectored ZIKV vaccines to support further clinical assessments.

[1] The Jenner Institute, Nuffield Department of Medicine, University of Oxford, The Henry Wellcome Building for Molecular Physiology, Roosevelt Drive, Oxford OX3 7BN, UK. [2] Center for Virology and Vaccine Research, Beth Israel Deaconess Medical Center, Harvard Medical School, Boston, MA 02215, USA. [3] Division of Immunology and Inflammation, Department of Medicine, Hammersmith Campus, Imperial College London, London W12 0NN, UK. [4] Nuffield Department of Medicine and Oxford NIHR Biomedical Research Centre, University of Oxford, Oxford OX3 7FZ, UK. [5] MRC-University of Glasgow Centre for Virus Research, University of Glasgow, Glasgow, G61 1QH Scotland, UK. [6] Aggeu Magalhães Institute, Oswaldo Cruz Foundation, 50670-465 Recife, Brazil. [7] The Jenner Institute, Nuffield Department of Medicine, University of Oxford, Old Road Campus Research Building, Roosevelt Drive, Oxford OX3 7DQ, UK. [8] Dengue Hemorrhagic Fever Research Unit, Office for Research and Development, Faculty of Medicine Siriraj Hospital, Mahidol University, Bangkok 10700, Thailand. [9] Division of Medical Sciences, John Radcliffe Hospital, University of Oxford, Oxford OX3 9DU, UK. Correspondence and requests for materials should be addressed to A.R.-S. (email: arturo.reyes@ndm.ox.ac.uk)

Zika virus (ZIKV) has caused a remarkably explosive outbreak in the Americas, with rapid spread to more than 70 countries[1]. Its association of ZIKV with Guillain-Barré syndrome[2], microcephaly in newborns[3] and person-to-person transmission[4] underlines the need for an efficacious vaccine that provides long-lasting anti-ZIKV immunity. Currently, no licensed Zika vaccines or antivirals are available to prevent or treat infection. However, exceptional progress has been made since the World Health Organization declared ZIKV a public health emergency, notably the description of the ZIKV atomic structure[5,6], the expansion of sequenced genomes of several ZIKV isolates and in particular the development of pre-clinical ZIKV challenge models in mice[7,8] and in non-human primates[9,10] These models have proven useful to explore efficacy of many ZIKV vaccine developments and many vaccine platforms such as subunit envelope protein, virus-like particles (VLP), live attenuated virus, inactivated virus, naked DNA vaccines, liposome-encapsulated RNA vaccines and viral vectored-based vaccines[11–15]. Equally important, attempts to optimise antigen secretion and presentation to immune cells include (1) addition or replacement of ZIKV envelope transmembrane domain to that of other flavivirus, (2) a great variety of different signal sequences to improve antigen secretion and (3) addition or replacement of ZIKV prM for those of other flaviviruses. However, it is unknown to which extent these subtle and yet non-unified modifications may play a role at maintaining long-lasting anti-ZIKV immunity.

In the present study we utilise the clinically validated replication-deficient chimpanzee adenovirus vector (ChAdOx1) as vaccine platform to express the *pre-membrane* (*prM*) and *envelope* genes of ZIKV. We demonstrate that immune responses vary upon the modulation of membrane anchors, in particular the transmembrane (TM) domain of the envelope and prM. The ChAdOx1 ZIKV vaccine candidates substantially reduce levels of viraemia in a challenge with a Brazilian ZIKV isolate in mice, with highest efficacy resulting in viral vectors expressing the ZIKV prM and envelope with deletion of the TM domain. Importantly, no evidence of in vitro antibody-dependent enhancement (ADE) to dengue is identified. Owing to the good safety and immunogenicity profile of the ChAdOx1 platform in humans and their suitability for high-scale production under Good Manufacturing Practices (GMP), the ChAdOx1 ZIKV vaccine is a robust candidate for further clinical assessment.

## Results

### Antigen design in DNA and ChAdOx1 ZIKV vaccines.
Because the Asian lineage of ZIKV is circulating in the Americas, efforts have focused on developing an Asian lineage-based vaccine[7,9,10]. By November 2015, before the World Health Organization declared ZIKV an emerging threat, only 5 complete genomes of Asian lineage were available (Fig. 1a). A year later, at least 47 Asian ZIKV sequences were deposited in the GenBank (Supplementary Fig. 1). Here we used a consensus-based approach to design cassettes for DNA vaccines and chimpanzee adenoviral vectored vaccines carrying Asian ZIKV sequences. Percentage identity of our consensus sequence versus ZIKV genomes was calculated (Fig. 1b) and a gene cassette was synthesised containing the *prM* followed by the *envelope* (E) transgene (prME) using the Asian consensus sequence (Fig. 1c). We designed modified versions of this cassette comprising *envelope* with inclusion or deletion of upstream *prM* (prME or E, respectively) and addition or deletion of the nucleotides encoding the envelope C-terminal transmembrane domain, a.a. 729–794 (ΔTM) of ZIKV (prME ΔTM or Env ΔTM; Fig. 1c); and cloned them into DNA vaccine plasmids and into ChAdOx1 vaccine platforms.

The expected size of the cloned cassettes was verified (Fig. 1d) and ZIKV envelope expression in mammalian cells was confirmed by western immunoblotting (Fig. 1e). We developed four DNA-based vaccines and four replication-deficient chimpanzee adenoviral vectored ZIKV vaccines (ChAdOx1 ZIKV).

### ZIKV immune responses after vaccination.
We performed immunogenicity assays to identify the most immunogenic candidates (Fig. 2). BALB/c mice immunised with a DNA vaccine using a homologous Prime-Boost (P-B) regimen (Fig. 2a, left) showed modest anti-ZIKV envelope antibody responses upon both prime and boost (Fig. 2b, left). In contrast, BALB/c mice immunised with a single, non-adjuvanted vaccine dose of ChAdOx1 (Fig. 2a, right) showed anti-ZIKV envelope antibodies at 2 weeks and peaked at 4 weeks (Fig. 2b, right). Interestingly, our results indicated that the concomitant inclusion of prM and ΔTM deletion elicited the highest anti-ZIKV envelope antibody titres peaking at 4 months post immunisation (Fig. 2b, right). Importantly, the durability of the anti-ZIKV antibody responses was maintained in the prME ΔTM-immunised group for up to 9 months, whilst lower antibody titres were observed in groups receiving the other vaccine candidates (Fig. 2b, right). Vaccine immunogenicity was also assessed in CD1 mice, confirming the capacity of the ChAdOx1 vaccines to induce antibody responses in outbred animals, which was sustained at high levels of antibody titres over a year, elicited by the same single and non-adjuvanted dose of ChAdOx1 prME ΔTM (Supplementary Fig. 3a).

T-cell responses were quantified by ex vivo interferon-γ (IFNγ) ELISPOT. BALB/c mice receiving DNA vaccines showed modest T-cell frequencies after stimulation with a pool of ZIKV envelope peptides both, at 2 weeks each after prime and boost, with the Env ΔTM group being the most efficient T-cell inducer (Fig. 2c, left). In contrast, a single immunisation of ChAdOx1 ZIKV vaccines induced far more robust T-cell responses than DNA vaccines as early as 2 weeks after prime and remained detectable for at least 3 months (Fig. 2c, right). Overall, T-cell responses to ChAdOx1 vaccines expressing Env, Env ΔTM and their prM counterparts were similar at week 2, with a trend by ChAdOx1 prME towards induction of lower T-cell responses. Interestingly, the prME ΔTM candidate maintained sustained responses at 2 weeks and 3 months after immunisation, indicating its ability to induce a good T-cell memory response.

### Efficacy of ChAdOx1 ZIKV vaccines in a ZIKV challenge model.
We used an Asian ZIKV strain to challenge mice[7] for the assessment of efficacy of the various ChAdOx1 ZIKV vaccines. BALB/c mice were immunised with non-adjuvanted ChAdOx1 vaccines and challenged 4 weeks later with 100 plaque-forming units (PFU) of a ZIKV isolated from Brazil (ZIKV-BR). Vaccine efficacy was measured in blood by ZIKV viral load (VL) using quantitative reverse transcriptionPCR (RT-qPCR) at several time points during 7 days (Fig. 3a). Both naive and ChAdOx1 control groups (Supplementary Fig. 3b) displayed the typical onset of viraemia with a well-established peak at d3, and all infected mice were able to clear ZIKV by d7[7] (Fig. 3b, naive). Mice vaccinated with the rest of the vaccine candidates (prME, Env and Env ΔTM) substantially decreased VL but did not afford sterile protection to all animals in those groups. Importantly, the ChAdOx1 prME ΔTM vaccine conferred complete protection against ZIKV-BR challenge with no detectable viraemia at any time point (Fig. 3b). Additional parameters reflecting vaccine efficacy were assessed, such as an analysis of 50-fold reduction in viraemia since day 1 after ZIKV challenge and reduction of viraemia peak at day 3 (Fig. 3c). These parameters can inform on the various capacity levels of ChAdOx1 ZIKV vaccines to confer

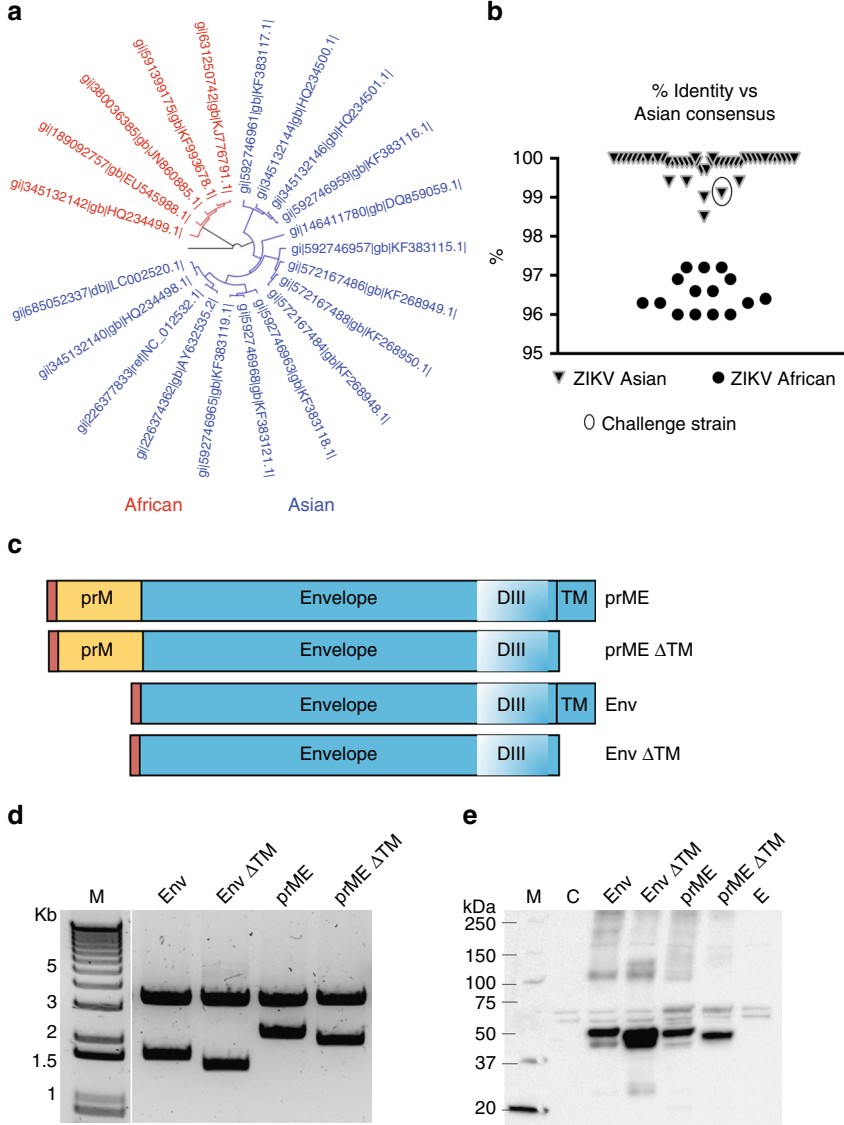

**Fig. 1** Zika vaccine design. **a** A phylogenetic tree for ZIKV genomes up to November 2015; blue and red labels represent the African and Asian lineages of ZIKV, respectively. **b** Conservation homology of an Asian consensus sequence versus all genomic sequences depicted in Supplementary Fig. 1. The circle represents the ZIKV-BR strain used for the challenge experiment. **c** Schematic representation of the Zika immunogen versions used in this study; red box represents the tPA leading sequence. **d** Restriction enzyme analysis of the plasmid DNA vaccine constructs that were further cloned into ChAdOx1 vector; a 3.3 kb band size represents the DNA vaccine plasmid back bone. **e** Expression of ZIKV immunogens by western immunoblotting using an anti-ZIKV envelope antibody. M molecular marker, C control, E empty plasmid. Uncropped images are shown in Supplementary Fig. 2

protection in this challenge model, indicating that the ChAdOx1 prME ΔTM is superior to the other vaccine candidates.

As in previous reports, anti-ZIKV envelope antibodies prior to the mouse challenge were quantified by enzyme-linked immunosorbent assay (ELISA) and reciprocal titres were calculated for each vaccinated group (Supplementary Fig. 4). This provided confirmation under different laboratory conditions and in a blinded ZIKV challenge study that the deletion of TM played an important role in the production of high antibody titres able to bind the ZIKV structural antigen. A comparison of the levels of ZIKV envelope antibody titres against the ZIKV-BR VL from all animals and all vaccine candidates under the study (Fig. 3d) revealed that an endpoint titre ≥ 3 was indicative of the strongest reduction of VL after the ZIKV-BR challenge (Fig. 3d, left), whereas prME and Env (Fig. 3d, right) presented endpoint titres < 3 that were related to a partial vaccine protection, although with a substantial decrease of VL of several log fold.

**ZIKV antigen secretion by membrane-anchoring components.** Although the anti-envelope antibodies elicited by ChAdOx1 prME were detectable, those induced by the counterpart vaccine ChAdOx1 prME ΔTM were significantly higher (Fig. 3e, left). Similar findings were observed in the ChAdOx1 Env ΔTM vaccine with greater antibody-binding activity to ZIKV envelope when compared to the ChAdOx1 Env (Fig. 3e, right). These data suggest that presence of the envelope transmembrane domain impairs antibody production elicited by our adenoviral vector. In an attempt to explain the differential immunogenicity among the ChAdOx1 ZIKV vaccine candidates, the subcellular localisation of the endogenously expressed vaccine antigen was explored by immunofluorescence (IF) using a monoclonal anti-flavivirus antibody 48 h after transfection (Fig. 3f). We detected differential staining with brighter positive signals across the cytoplasmic compartment in the vaccines expressing prME and Env, when compared to their counterparts prME ΔTM, that showed positive signal mainly

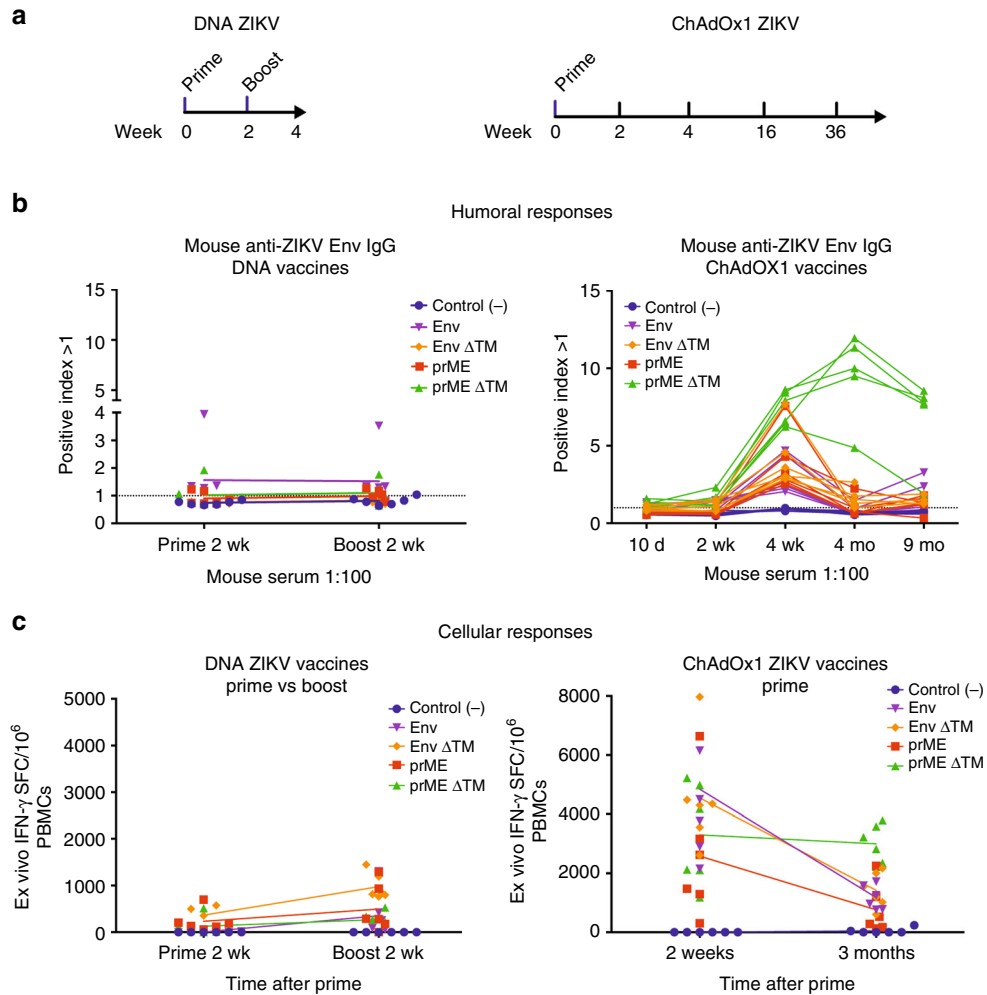

**Fig. 2** Immune responses elicited by DNA versus ChAdOx1 vaccines. **a** For ZIKV DNA vaccines, BALB/c mice ($n = 6$/group) were immunised intramuscularly (i.m.) with a dose of 100 µg/mouse, followed by a DNA Boost 2 weeks thereafter. For ChAdOx1 ZIKV vaccines, a single dose of $10^8$ IU/mice was i.m. administered. Blood samples were obtained at several time points for either ELISA or ELISPOT assay. **b** Humoral responses elicited by DNA Prime and Prime-Boost after 2 weeks (left) and by a single immunisation of ChAdOx1 ZIKV vaccines at 10 days, and 2, 4, 16 and 36 weeks (right). Antibody responses were quantified by ELISA with plates coated with ZIKV envelope protein. Positive index was calculated as depicted by ELISA kit manufacturers. **c** PBMCs–IFNγ-producing cells from DNA Prime-Boost after 2 weeks (left) and ChAdOx1 ZIKV vaccines at 2 weeks and 3 months after single immunisation were quantified by ELISPOT. 20-mer peptides spanning the ZIKV envelope protein (10 µg/ml) were used for stimulation. Line colours and shapes represent mice vaccinated with each vaccine. Antibody production by ELISA and cellular responses by ELISPOT were replicated three times in the laboratory. A positive control sample was included to validate the ELISA assay

around the nucleus and with punctuate patterns across the cytoplasmic compartment and for Env ΔTM, with less bright staining around the nucleus. Furthermore, we have assessed the presence of the ZIKV envelope antigen in transfected cells in both, the intracellular compartment and the supernatant (Fig. 3g and Supplementary Figs. 5, 9 and 10). HEK-293 cells expressing prME ΔTM and Env ΔTM antigens secreted ZIKV envelope to the culture supernatant with higher efficiency than their counterparts Env and prME, with prME ΔTM secreting with highest efficiency in vitro (Fig. 3g, supernatant fraction, red arrows), while having the lowest levels of envelope in the cell fractions. For prME, we detected ZIKV envelope only in concentrated supernatant at 40 h but not at 24 h after transfection, whereas the Env construct containing the TM domain did not support secretion of the antigen to the supernatant at these time points (Supplementary Figs. 9, 11). These results suggest that antigen secretion in our adenoviral vaccine platform is modulated by membrane-anchoring components of the ZIKV structural proteins.

**ZIKV-neutralising capacity in vaccinated mice sera**. We tested BALB/c mice sera obtained 4 and 16 weeks after a single ChAdOx1 ZIKV vaccination to assess ZIKV antibody neutralisation activity in vitro. The method we used is based on the high-throughput microneutralisation (MN) assay described in several studies that have evaluated the efficacy of experimental vaccines against ZIKV in mice and macaques[9,16–18], as well as in the validated neutralisation assay used for a phase 1 clinical zika DNA vaccine[7,19], and recently in three phase I clinical trials of a purified inactivated ZIKV vaccine[20], with small modifications. Briefly, 50–100 PFU of the Brazilian ZIKV-PE243[21] strain previously mixed with different threefold dilutions of mouse serum (1:20–1:43 740) are incubated with Vero cells in a 96-well format. After a 3-day incubation, cells are lysed and the levels of ZIKV envelope protein from infected cells are quantified by measurement of optical density using a highly sensitive sandwich ELISA format. For the sandwich ELISA, a rabbit antiserum R34 was developed to recognise a region between a.a. 147 and 161 that

is unique to each flavivirus and that forms a conformational exposed loop (Supplementary Fig. 6).

At 4 weeks post vaccination, sera from the ChAdOx1 prME and ChAdOx1 prME ΔTM-vaccinated group showed neutralisation activity against ZIKV-PE243 with MN50s ranging from 114 to 432 and from 292 to 434, respectively (Fig. 4a). In comparison, at 16 weeks after the ChAdOx1 ZIKV vaccination, MN50s were surprisingly maintained at high levels in the prME ΔTM group with MN50s ranging from 253 to 1600, while the ChAdOx1 prME showed lower MN50s of 30 to 89. These results indicate a beneficial effect of including the prM region as part of an adenoviral vectored vaccine and it highlights the importance of

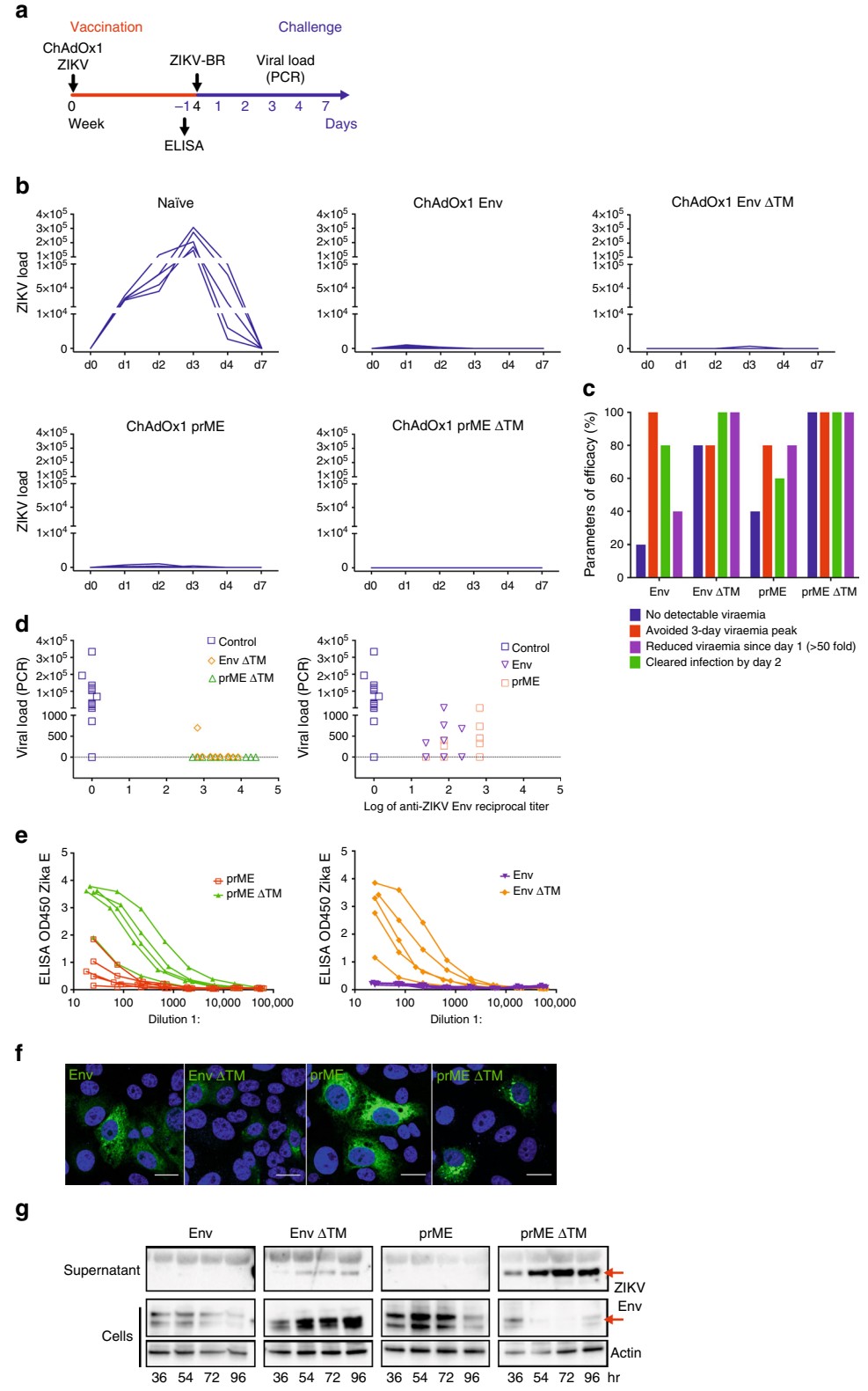

deleting the 3′ TM region of the envelope to maintain good memory responses. Virus neutralisation was also assessed in outbred CD1 4 months after a ChAdOx1 ZIKV vaccination (Supplementary Fig. 3c), showing that ChAdOx1 prME ΔTM vaccine is able to induce MN50 values of 113 to 150, whereas ChAdOx1 prME group showed MN50 values of 10.38 to 18.23. Taken together, this confirms that ΔTM is an important factor to maintain neutralising antibody activity over longer periods of time.

**Lack of evidence of ADE to dengue 2 after ZIKV vaccination.** Next, we tested whether antibodies induced by ChAdOx1 prME ΔTM vaccine cross-react with the envelope protein from all dengue virus (DENV) serotypes (DENV1-4), by ELISA. Our results indicated that ZIKV antibodies are able to bind DENV (Fig. 4b). We then addressed the possibility of ADE to DENV infection being induced by ZIKV vaccination, using a well-established in vitro assay for ADE to DENV2[22] (Fig. 4c), as it is the most difficult serotype to protect against with the only licensed dengue vaccine[23,24] and the most suitable serotype to address ADE in vitro. We assessed the ability of sera from ChAdOx1 ZIKV-vaccinated mice groups to promote ADE in the human myeloid cell line U937, which is less permissive to infection by DENV in the absence of ADE[22]. DENV2 was pre-incubated with a titration of mouse sera at 4 weeks after a single immunisation with ChAdOx1 ZIKV vaccines, and the DENV2-serum complex was used to infect U937 cells. As positive controls, we used anti-DENV2 antibodies that leads to in vitro ADE by DENV2 (Fig. 4c, top left and Supplementary Fig. 8) and as a negative control we used sera from the ChAdOx1 control vaccine (Fig. 4c, top right). None of the sera obtained from ChAdOx1-vaccinated groups induced enhancement of DENV2 infection regardless presence or absence of TM, whereas there was a 50-fold ADE increase when using an anti-DENV2 antibody (Fig. 4c, bottom left and right). These results demonstrate that antibodies raised to ZIKV in ChAdOx1 ZIKV-vaccinated mice do not promote ADE of DENV2 in vitro despite cross-reacting with envelope of all DENV serotypes.

## Discussion

We have developed a ChAdOx1 ZIKV vaccine with the ability to provide 100% protective efficacy against a ZIKV-BR isolate. In our hands, the addition of prM and deletion of TM from ZIKV envelope are important factors in driving improved protective immunity, while promoting long-term memory responses by enhancing and maintaining antibody production as well as T-cell responses, when compared to previously reported vaccines. Our vaccine design is different from the previously published ZIKV vaccines[7,9,10]. Larocca et al.[7] and Abbink et al.[9] showed that addition of the TM from envelope was important to elicit full protective efficacy in vaccines carrying a

Japanese encephalitis virus (JEV) leader sequence and a 93 a.a. deletion of pr from prM. Importantly, the impact of ΔTM in the immunogenicity and efficacy was not assessed in the adenoviral platform RhAd52-prM-Env, only in DNA-based vaccines. We found that DNA vaccines showed differential immunogenicity when compared to the adenoviral counterparts and hence the importance of constructing four adenoviral vaccine candidates and assessing their protective capabilities. Dowd et al.[10] developed a DNA vaccine consisting of a JEV leader sequence followed by a full prM and the full ZIKV envelope; or with a replacement of the endogenous stem and TM from ZIKV with those from JEV, however the authors did not assess the impact of the ΔTM. Richner et al. have developed an mRNA-based vaccine in lipid nanoparticles encoding the full prME of ZIKV and using a leader sequence from the human *IgE* gene. Results from this mRNA-based vaccine show that two doses elicited strong and protective antibody responses[15]. Alternatively, Pardi et al.[13] achieved neutralising antibodies in a single intradermal dose with an mRNA-based nanoparticle vaccine, this time using the full prME from ZIKV, including a leader sequence from major histocompatibility complex (MHC) class II. However, neither of the mRNA-based ZIKV vaccine studies assessed the impact of a ΔTM deletion in their candidates to study its differential immunogenicity. On the other hand, Kim et al.[25] have reported a protective two-dose regimen with a human adenoviral vaccine (Ad5) encoding the prME of ZIKV with a deletion of envelope transmembrane domain, being replaced with a T4 fibritin foldon trimerization domain, as well as a human leader sequence, both to improve antigen secretion.

It has been shown that an optimal leader sequence is protein-dependent[26] and differences observed in these studies may be also attributed to the use of alternative leader sequences (such as those from JEV, MHCII and IgE). We have used a non-viral leader sequence (tissue plasminogen activator; tPA)[27], which has been used safely in humans for many vectored vaccines[28] and which is unlikely to induce responses in humans[29]. Regarding previous vaccine developments, it is important to take into account pre-existing humoral and cellular cross-reactive immune responses to human adenovirus; as well as when using JEV sequences to improve antigen secretion (leader sequence and/or JEV trans-membrane domains), as they might have a detrimental effect on the efficient ZIKV antigen production, i.e., Asian populations exposed to or vaccinated against JEV. Moreover, different vaccine platforms such as DNA, mRNA-based vaccines, VLPs or adenoviral vectors, ZIKV strains used for the immunogen design, as well as different routes of administration may also account for different immunogenicity and efficacy levels achieved. In this study, we have used a non-lethal challenge model in BALB/c mice and we have measured vaccine efficacy by means of partial or complete reduction of viraemia by PCR. However, we do not rule out the possibility of ZIKV presence in key organs such as brain, liver, spleen, reproductive tract, etc. Further studies, such as the

---

**Fig. 3** Assessment of protective efficacy induced by ChAdOx1 ZIKV vaccines. **a** BALB/c mice (*n* = 5) immunised with a single i.m. injection of ChAdOx1 ZIKV vaccines and naive mice were intravenously challenged with 10⁵ VP of ZIKA-BR at week 4 after prime. Pre-challenge serum samples were collected for ELISA as shown. **b** Viral load in ZIKV-challenged groups was monitored for 7 days in sera to follow the onset of viraemia. Each blue line represents one mouse. **c** Assessment of various parameters of viraemia in groups vaccinated with ChAdOx1 ZIKV vaccines (100% = 6 out of 6 mice). **d** Pre-challenge reciprocal ELISA titres (Supplementary Fig. 4) were plotted against global ZIKV VL after ZIKV challenge and a comparison was made between ChAdOx1 Env ΔTM versus ChAdOx1 prME ΔTM (left) and ChAdOx1 prME versus ChAdOx1 Env (right). **e** ELISA OD 450 reads of serially diluted serum samples were plotted to assess the impact of ΔTM in antibody production. Antibody production by ELISA in pre-challenge sera was replicated two times. **f** Immunofluorescence analysis of Vero cells expressing the ZIKV immunogens (green) to assess subcellular staining and distribution. Antigen was detected using a commercial anti-flavivirus antibody. Blue (DAPI) represents the nucleus and scale bar is 20 μm. Cell transfections were performed in technical duplicates in two biological replicates. **g** Kinetics of ZIKV envelope antigen expression in HEK-293 cells by western blot. Antigen was detected using an anti-ZIKV monoclonal antibody in both, non-concentrated supernatant and cell extracts. Red arrows, ZIKV envelope. Actin was used as a loading control. Expression of antigen was verified in three biological replicates. Uncropped blots are shown in Supplementary Fig. 5

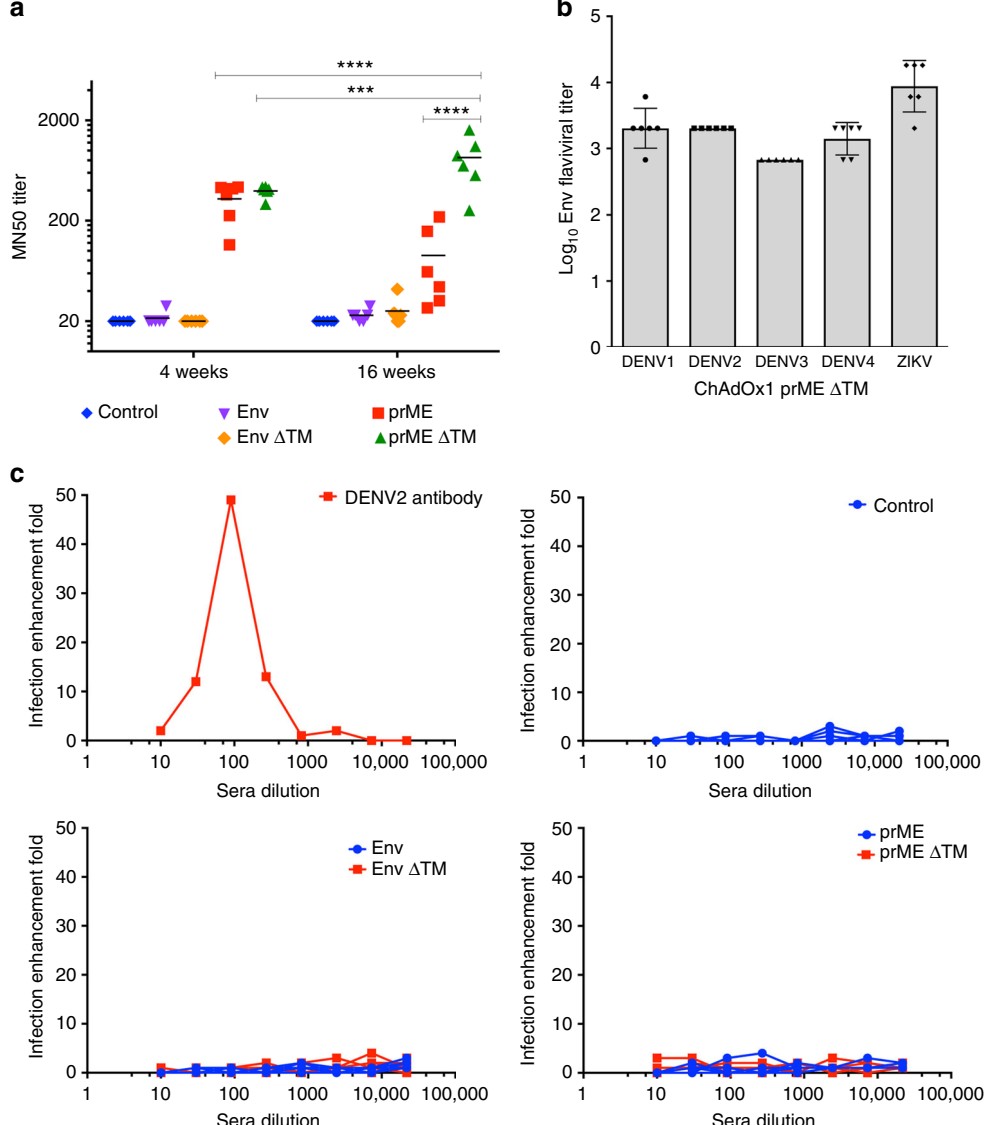

**Fig. 4** ZIKV neutralisation assay and antibody-dependent enhancement (ADE) to DENV2 infection. **a** Microneutralisation titre 50 (MN50) assay in Vero cells was assessed upon incubation of ZIKV-BR in the presence of serially diluted sera from BALB/c mice vaccinated with ChAdOx1 ZIKV vaccines at 4 weeks and 4 months. Results show the average of three biological replicates, with duplicates and black lines represents the mean ($n = 6$). A two-way ANOVA followed by Sidak's Multiple comparison was performed. ****$p < 0.0001$, ***$p < 0.0003$. Supplementary Table 1 shows the multiple comparison between all the groups. **b** ELISA cross-reactivity of DENV1-4 in sera from mice vaccinated with the highly protective ChAdOx1 prME ΔTM vaccine. Bars represent the average and error is the s.d. **c** Infection of U937 cells by a DENV2 strain in the presence of sera from mice vaccinated with ChAdOx1 ZIKV candidates. Infection enhancement fold was calculated as the ratio of focus-forming units (FFU) of an anti-DENV2 monoclonal antibody to that of the FFU in the presence of serial dilutions of mouse sera collected 4 weeks after ChAdOx1 ZIKV vaccination. Each coloured line and shape represents a mouse for each group. $n = 6$/vaccinated group. Data represent three technical replicates, with a positive control anti-DENV2 antibody

use of a highly susceptible ZIKV challenge model in A129 mice, are needed to explore ZIKV presence in those organs after a single and non-adjuvanted ChAdOx1 ZIKV vaccination. Another correlate of vaccine efficacy is assessing the ZIKV neutralisation of sera from vaccinated mice in vitro. For this we have used a MN assay, highly similar to that performed in phase I clinical trials of several Zika vaccines in the pipeline[20,30,31]. Furthermore, MN is a validated assay used previously in several publications to assess ZIKV vaccine efficacy in mice and non-human primates[9,16–18], as well as other flaviviruses[19]. Other neutralisation assays, such as traditional PRNT50, FRNT50 and reporter viral particle assay (using green fluorescent protein, Luciferase and mCherry as reporters) have also proven to be suitable. However,

it is challenging to perform an accurate comparison of vaccine efficacy between other reported vaccine candidates, as neutralisation 50 (NT50) titres vary from one assay to another (different ZIKV strains, different ZIKV purification methodologies and different cell lines [i.e., Vero and BHK-2] and different detection methods). Nonetheless, we have performed a fair head-to-head comparison between our high-throughput MN assay versus a traditional PRNT assay. Our results demonstrate that our MN assay is more stringent at detecting ZIKV infection than PRNT (Supplementary Fig. 3d). Therefore, the MN50 values in our study are moderate when compared to PRNT. Similar observation has been reported by Dowd et al.[10] in which MN50 titres are actually lower than those of a parallel FRNT50 assay.

However, MN assays together with ZIKV challenge models are both highly informative to efficiently down-select Zika vaccine candidates in the preclinical setting.

The risk of inducing ADE[32], a process in which vaccine- or infection-induced cross-reactive antibodies facilitate DENV infections[33] by promoting viral entry via Fcγ receptors, adds another level of complexity in the design of a safe and efficacious ZIKV vaccine. ZIKV and DENV co-circulate in endemic regions making ADE a concern as it is not known whether antibodies elicited by ZIKV vaccines may in turn exacerbate other flaviviral infections such as dengue. A protective vaccine that does not induce ADE while providing long-lasting immunity to ZIKV would be a major breakthrough to halt the epidemics. Here we demonstrate that anti-ZIKV antibodies did not induce ADE to DENV2 upon immunisation with ChAdOx1 ZIKV vaccines in vitro. In this study, we incubated mouse sera with ZIKV and then inoculated this mixture onto human U937 cells expressing a human Fc receptor. Although we have reacted the mouse monoclonal antibody (mAb; 4G2) with DENV2 and observed the ADE phenomena in human U937 cells (Supplementary Fig. 7), it is possible that mouse Fc could present a degree of incompatibility with the human Fc receptor in U937 cells. Additional in vivo experiments should be carried out to completely rule out any enhancing effect of anti-ZIKV antibodies to DENV. In vitro ADE to ZIKA virus has been demonstrated in human sera positive for DENV that cross-reacted with ZIKV[22]. Richner et al.[15] have also shown that a mutation of the fusion loop of ZIKV envelope decreased the risk of ADE to DENV2 in their mRNA-based vaccines in vitro. A recent study, interestingly, shows that DENV antibodies did not increase viremia and disease after a ZIKV challenge in macaques, although ADE to ZIKV was detected in vitro[34]. Similarly, cross-reactive human mAbs by ZIKV or DENV directed primarily and cross-reactive to EDI/II can mediate heterologous ADE, while in vivo observations indicate that the same cross-reactive anti-ZIKV mAbs can lead to ADE and enhance infection by DENV, while the opposite effect for ZIKV ADE does not take place[35]. Therefore, assessing ADE both in vitro and in vivo should be taken into account and mechanistically explored in current and ongoing ZIKV vaccine developments.

Chimpanzee adenovirus vaccine platforms such as ChAdOx1 offer a good safety profile in phase I clinical trials in humans[36–40]; efficient biomanufacture of millions of doses of non-replicating adenoviral vectors is possible using GMP-approved cell lines. They avoid pre-existing immunity to human adenoviruses, and a single non-adjuvanted dose of ChAdOx1 is enough to achieve remarkable breadth, durability and potency of both humoral and cellular immune responses. A ChAdOx1 ZIKV vaccine engineered to positively impact immune responses over longer periods of time, by means of deletion of membrane anchors, is a valid platform to be explored in further clinical trials.

## Methods

**Animals.** Female inbred BALB/c (H-2d) and outbred CD1 (ICR) mice (6–8 weeks) were used for the assessment of immunogenicity (n = 6 mice/group). ZIKV challenge was performed in BALB/c mice (n = 5/group). Mice were purchased from Envigo (UK) and from Charles River (USA). The experimental design took into account the 3R reduction (replacement, reduction and refinement). No randomisation was used in this work.

**Immunisation of mice.** For DNA vaccines encoding the ZIKV virus antigens, mice received a prime of 100 μg of DNA vaccine/mouse in endotoxin-free phosphate-buffered saline (PBS); 2 weeks after the prime a boost injection was preformed using the same concentration of DNA vaccine. For adenoviral vaccines, mice were vaccinated with a single dose of chimpanzee adenovirus vectored vaccine (ChAdOx1) encoding the ZIKV antigens at a dose of 1 × 10^8 infectious units. All DNA vaccines and viral vector vaccines were administered intramuscularly and diluted

in endotoxin-free PBS. Empty DNA vaccine and a ChAdOx1 expressing an unrelated malarial antigen (cCSP) were used as control vaccines.

**Ethics statement.** All animals and procedures were used in accordance with the terms of the UK Home Office Animals Act Project License. Procedures were approved by the University of Oxford Animal Care and Ethical Review Committee (PPL 30/2414). Animal experiments performed at Harvard were approved by the BIDMC Institutional Animal Care and Use Committee.

**Bioinformatics analysis of ZIKV genomes and proteins.** Complete ZIKV genomes deposited in the GenBank in November 2015 (no less than 10.5 kb) were used to construct a phylogenetic tree, which allowed us to further identify and design our vaccine transgene candidates. Further phylogenetic trees were built for genome sequences deposited in the gene bank between April and October 2016. Alignment analysis for both nucleotides and protein sequences, consensus analysis and phylogenetic trees were produced by focusing only in the structural and non-structural coding sequences. Geneious software was used to generate the phylogeny trees using a Neighbour-Joining method. Sequence alignments between zika, dengue and yellow fever proteins allowed us to annotate the ZIKV consensus sequences generated in November 2015 using the following features (start codon, transmembrane domains and known flaviviruses enzymatic cleavage sites).

**Transgene design.** For the Asian Lineage vaccine design, prM and envelope coding sequence (translated to 670 a.a.) from the Asian ZIKV consensus sequence was codon optimised, and a tPA leading nucleotide sequence was added before the 5′ end of prM. To improve initiation of translation a Kozak consensus sequence was included before the 5′ end of tPA. Finally, the transgene design included the required enzymatic restriction sites to allow the in-frame cloning of the transgene between the cytomegalovirus (CMV) promoter and the polyA sequence region contained in our shuttle and expression vector (pMono). Synthetic gene cassette was produced by GeneArt® and was named prME.

The GeneArt® prME plasmid was used as template to further generate the following versions by PCR subcloning, including restriction sites, stop codons or other features as follows:
-prME with the TM domain of envelope deleted (prME ΔTM):
Forward primer AGAAGAGGATCGAAGCTTGCCATCACCAGAAGAGGCAGCG
Reverse primer GCATGCTCGAGCGGCCGCTCATCACATTCTCTTGGCTCCCCGC
-Full envelope (Env):
Forward primer GAAACCGGTAAGCTTATGCGGTGTATCGGCGTGTCC
Reverse primer ACCGGTCAGACGTACTCGA
-Envelope with TM deleted (Env ΔTM):
Forward primer GAAACCGGTAAGCTTATGCGGTGTATCGGCGTGTCC
Reverse primer CACCAATGCGGCCGCCTCGAGCTATTACAGAGAGTTCAGGGCTCCG

**DNA vaccine production.** The prME and prME ΔTM were digested with KpnI and NotI to allow in-frame ligation between the CMV promoter and the poly(A) regions contained in the DNA vaccine plasmid (pMono). The remaining PCR-subcloning transgenes (Env and Env ΔTM) were digested with AgeI and NotI to allow in-frame ligation between the tPA leader sequence and the poly(A) regions of pMono. pMono was digested accordingly to generate the following DNA vaccines: pMono prME; pMono prME ΔTM; pMono Env; and pMono Env ΔTM. Zika DNA plasmid vaccines were expanded in Escherichia coli and Endo-free Giga Prep (Qiagen) was used for plasmid purifications. DNA vaccines were verified by restriction analysis and 5′ and 3′ flanking sequencing. In addition, DNA vaccines were transfected in HEK-293 and Vero cells to verify the correct expression of the transgenes.

**Adenoviral Zika vaccine production.** All of the DNA constructs required for ChAdOx1 were cloned in two steps. In the first step, unique KpnI and NotI sites were used to insert the synthetic transgenes into an adenovirus entry plasmid. The transgene was placed upstream of BGH poly(A) transcription termination sequence and under the control of the CMV promoter. The entry plasmid containing attL regions sequences was recombined with those attR regions contained in the destination vector ChAdOx1 using an in vitro Gateway reaction (LR Clonase II system, Invitrogen). Successfully recombined ChAdOx1 ZIKV plasmids were verified by DNA sequencing using flanking primers (promoter forward primer and poly-(A) reverse primer). Standard cell biology and virology techniques were performed to generate the non-replicative adenoviral vectors[36].

**Cell culture.** HEK-293 and Vero cells (ATCC®, CRL-1573™ and ATCC CCL81™, respectively) were grown in Dulbecco's modified Eagle's media (DMEM) supplemented with 10% fetal bovine serum, 1% L-glutamine and 1% non-essential amino acids. Cell number and viability were calculated by trypan blue

staining using the Countess Automated Cell Counter (Life Technologies). Both cell lines are described in the NCBI Biosample database.

**Immunofluorescence microscopy.** Transfected Vero cells grown on coverslips were fixed in 4% paraformaldehyde in 1× PBS (10 min, room temperature), permeabilised in 0.1% PBS-Triton X-100 for 20 min at room temperature (RT) and blocked in 2.5% goat serum/0.5% bovine serum albumin (BSA) in PBS for 45 min at RT. The primary antibody (anti-flavivirus group antigen D1-4G2-4-15, 1/250, Merck Millipore) was diluted in 1% BSA in PBS and incubated with samples for 1 h at RT. After successive washes in PBS, secondary antibody coupled to Alexa Fluor 488 was added for 1 h at RT in the dark. Samples were washed in PBS with a last wash in distilled water and mounted in Prolong-DAPI mounting reagent (Life Technologies). Cells were acquired on a LSM 880 inverted confocal microscope using Zen 2.1 software (Zeiss). Images were analysed using ImageJ software (National Institutes of Health).

**Western blot analysis.** Cells were washed twice with ice-cold PBS and harvested in cold 1× PBS. 1× and 4× Laemmli buffer were added to cells and supernatants, respectively, and boiled at 100 °C for 5 min. Equal amounts of total cell extracts and 15 μl of cell supernatants were resolved by SDS/PAGE and transferred to polyvinylidene fluoride membranes. In some cases, supernatants were 10× concentrated using Amicon Ultra-0.5 ml centrifugal filters. Blots were blocked with 1× PBS-Tween-5% milk and incubated with anti-ZIKV envelope seropositive mice sera (1:500) or a mAb versus ZIKV envelope (AZ1176, Clone 0302156), Aalto Bio Reagents), followed by incubation with horseradish peroxidase (HRP)-conjugated secondary antibody (1:5000). Chemiluminescence (Perkin-Elmer Life Sciences, Boston, MA) was visualised using the BioRad ChemiDoc SRS device. Uncropped images of the blots are shown in Supplementary Figs. 2, 5, 9, 10 and 11).

**Whole IgG ELISA.** Recombivirus™ mouse anti-ZIKV envelope protein IgG ELISA kits (Alpha Diagnostic International, RV-403120-1) were used under the manufacturer protocol. Briefly, 96-well plates coated with ZIKV envelope protein were equilibrated with 300 μl of kit working wash buffer. Serial dilutions (threefold) of sera from vaccinated mice were added. Diluted sera were incubated at RT for 1 h, and after four times washing buffer incubations, 100 μl/well of anti-mouse IgG HRP conjugate working solution was added for 30 min at room temperature. Plates were washed five times and developed 15 min at room temperature with 100 μl of tetramethyl benzidine (TMB) substrate, then stopped by the addition of 100 μl of stop solution. Absorbance was measured at 450 nm on a microplate reader. ELISA optical densities (ODs) were compared between all vaccinated groups at different sera dilutions. Alternatively, a relative positive index was calculated following the manufacturer Recombivirus™ Mouse Anti-Zika virus ELISA kit. Briefly, positive index was calculated by the OD mean plus two times the s.d. of the control/naive sera. Then, we divided the OD of each sample by the positive index. Values below 1 are considered negative and values above 1 are considered positive. Relative positive index then was plot for every vaccinated mouse per group.

For pre-challenge ELISA, endpoint titres were defined as the highest reciprocal serum dilution that yielded an absorbance greater than twofold over background values, as calculate elsewhere[8–10].

DENV1-4 ELISA: Dengue Env-2 45 kDa (Den021), Dengue Env-1 45 kDa (Den022), Dengue Env-3 45 kDa (Den023) and Dengue Env-4 45 kDa (Den024) from Dundee Cell Products Ltd. were used to coat Maxisorp NUNC 96-well plates (2 μg/ml) and ELISAs performed to determine the reciprocal titre of the sera dilution.

**Ex vivo IFNγ ELISPOT assay.** ELISPOTs were carried out using peripheral blood mononuclear cells (PBMCs) isolated from the blood. Briefly, MAIP ELISPOT plates (Millipore) were coated an anti-mouse IFNγ mAb, after 1 h blocking with complete DMEM media (10% fetal calf serum). Isolated PBMCs (using ACK buffer solution) were plated alongside with 20-mer specific peptides overlapped by 10 a.a. (10 μg/ml) and $2.5 \times 10^5$ splenocytes from naive mice per well. After 16 h incubation, cells were discarded and plates washed with PBS. Following this, 50 μl of biotinylated anti-mouse IFNγ mAb (1:1000 in PBS) was added to each well and incubated for 2 h. After washing, plates were incubated with 50 μl of ALP (1:1000 in PBS) reagent for 1 h. After another washing step, developing solution (BIORAD) was used. Once spots were visible, the reaction was stopped by washing plate off with water. Spots were acquired using an ELISPOT reader. Spot-forming cells/$10^6$ PBMCs producing IFNγ were calculated.

**ZIKV challenge model.** ZIKV challenge was performed as described[7]. Briefly, naive and vaccinated BALB/c mice ($n = 5$/group) were infected at week 4 post immunisation by the intravenous route with $10^5$ viral particles ($10^2$ PFU) of ZIKV-BR. VLs following ZIKV challenge were quantitated by RT-PCR at days 1, 2, 3, 4 and 7. Sample size was determined to achieve 80% power to detect significant differences in protective efficacy. Vaccines were administered in a blind experiment. After the experimental outcome of the ZIKV challenge and the pre-challenge ELISAs, samples were decoded.

**Revese transcription-PCR.** VLs were assessed as depicted[7]. Briefly, RNA was extracted from serum with a QIAcube HT (Qiagen). RNA was purified using the RNA clean and concentrator kit (Zymo Research), and RNA quality and concentration was assessed by the BIDMC Molecular Core Facility of Harvard University. Log dilutions of the RNA standard were reverse-transcribed and included with each RT-PCR assay. VLs were calculated as virus particles/ml. Assay sensitivity was 100 copies/ml. The infectivity of virus in peripheral blood from ZIKV-challenged mice was confirmed by PFU assays. Specific primers were used to amplify a region contained in the Capsid of the ZIKV genome.

**Polyclonal anti-ZIKV E antiserum R34.** We identified a short region (a.a. 147–161) around the N154 glycosylation site of the envelope protein that is unique to each flavivirus and that forms an exposed loop (the so called 150 loop) on the surface of the envelope protein (Supplementary Fig. 6a, b), and then raised a rabbit antiserum (R34) to it which specifically recognised the ZIKV envelope protein (Supplementary Fig. 6c). A peptide corresponding to amino acid residues 144–166 of ZIKV envelope (HGSQHSGMIVNDTGHETDENRAK) with an additional C-terminal cysteine residue was synthesised by Isca Biochemicals (UK) and directionally conjugated to keyhole limpet haemocyanin (KLH) using maleimide chemistry. Two rabbits were immunised with the peptide-KLH conjugate combined with Freund's adjuvant, following a standard protocol. This work was carried out by Dundee Cell Products (UK). The immune sera were tested by western blot, ELISA and IF for specific reactivity with ZIKV E protein, and R34 was selected as the better antiserum. Total IgG was purified from R34 antiserum by protein G affinity chromatography.

**ZIKV MN assay.** Vero cells were seeded in 96-well plates at $7 \times 10^3$/well and incubated at 37 °C in 5% CO2 for 1 day to form a subconfluent monolayer. Aliquots of Brazilian ZIKV-PE243[21] at 100 PFU in 15 μl were mixed with an equal volume of twofold dilutions of sera in medium and incubated at 37 °C for 1 h. The cells were infected with the serum/virus mixture (30 μl/well), incubated at 37 °C for 1 h, and then 100 μl of medium was added to each well. At day 3 post infection, cells were lysed in lysis buffer (20 mM Tris-HCl [pH 7.4], 20 mM iodoacetamide, 150 mM NaCl, 1 mM EDTA, 0.5% Triton X-100 and Complete™ protease inhibitors) and the viral E protein quantitated by sandwich ELISA (below). The amount of envelope detected correlates with the level of virus infectivity. Values representing % of ZIKV infectivity relative to the control (i.e., virus not pre-incubated with immune sera) were plotted using Graphpad Prism 6, and nonlinear regression (curve fit) performed for the data points using Log (inhibitor) versus response (variable slope) to determine MN50 titres. The MN50 titre was defined as the serum dilution that neutralised >50% of ZIKV. This format resulted in a very robust MN assay, with a signal-to-noise ratio of >10 and a Z-factor of 0.8, which is optimal value for high-throughput screening assays. A comparison between our MN assay versus a traditional PRNT assay[41] was performed, using a series of twofold dilutions of a highly potent monoclonal neutralising antibody to ZIKV (EDE1 C8)[42].

**Sandwich ELISA to assess ZIKV infectivity.** Immulon 2 HB plate (Thermo Scientific™) 96-well plates were coated with 3 μg/well of purified pan-flavivirus MAb D1-4G2-4-15 (ATCC® HB112™) in PBS and incubated overnight at RT. The antibody was removed and the wells blocked for 2 h at RT with 2% skimmed milk powder in PBS containing 0.02% Tween-20 (PBST). After washing with PBST, ZIKV-infected cell lysates were added and incubated for 1 h at RT. Wells were washed with PBST, incubated with R34 IgG at 6 μg/ml in PBST for 1 h at RT and washed again. The antibodies bound to ZIKV envelope protein were detected using anti-rabbit HRP conjugate (Abcam) and TMB (Sigma) substrate. Absorbance at 450 nm was measured in a microplate reader.

**In vitro assessment of ADE.** Serially diluted mAb or mice sera samples were incubated with virus DENV2 strain 16681, Moi = 2 for 1 h at 37 °C before adding to U937 cells. After 3-day incubation, supernatants were harvested and viral titres determined by focus-forming units. Fold enhancement was calculated by comparison to viral titres in the presence or absence of antibody[22]. Antibodies used as positive control are mouse anti-Flavivirus envelope protein antibody (4G2) and anti-envelope dimer Epitope 1 [752-2C8].

**Statistics.** Graphing and statistical analysis were performed using GraphPad Prism version 7.00 for Mac, GraphPad Software, La Jolla California USA, www.graphpad.com. For pre-challenge ELISA, the analysis used was the one-way analysis of variance (ANOVA), followed by Turkey's multiple comparisons test. For the ZIKV neutralisation assay, a two-way ANOVA followed by Sidak's multiple comparison was performed. All graphs were plotted with bars and errors, representing the mean and the s.d. Assumptions of data are normal distribution, homogeneity, independent and groups must have same sample size.

**Data availability**. The authors declare that the data supporting the findings of this study are shown in the article and in the Supplementary section and are available from the authors upon request.

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

## Acknowledgements

We would like to thank the Jenner Institute's Vector Core Facility for producing and supplying the recombinant viral vectors. A.R.-S., A.V.S.H. and S.C.G. are Jenner Investigators and an Oxford Martin Fellows. R.D.L.N. and D.F.C. were supported by Capes, Facep, CNPq and LNCC. This report is independent research funded by the UK Department of Health and Social Care through Innovate UK "New vaccines for global epidemics: development and manufacture" grant No. 972216 (A.R.-S.), and also funded from an ODA budget (Global Health (ODA), 16/107/05—Design, development and GMP manufacture of a Zika vaccine) (A.H.P. and A.R-S). The views expressed in this publication are those of the author(s) and not necessarily those of the Department of Health and Social Care. We also acknowledge funding by the UK Medical Research Council [MC_UU_12014 (A.H.P. and A.K.) and MR/N017552/1 (A.K.)]. Juthathip Mongkolsapaya is supported by an MRC-Newton Fund grant, Gavin Screaton is a Wellcome Trust Senior Investigator.

## Author contributions

A.R.-S. and C.L.-C. directed the project and commissioned the work. C.L.C. designed, constructed and characterised the vaccines. C.L.-C., P.A. and R.A.L. designed and performed the animal experiments. M.B., A.B.-Z., P.A. and R.A.L. performed the ZIKV challenge model and the RT-PCR viral loads. C.L.-C., P.A. and R.A.L. performed ELISA assays and analysed ZIKV challenge data. W.D. and J.M. designed, performed and analysed in vitro ADE experiments. C.L.-C. performed ELISPOTs, cell culture, transfections and western blots. Y.C.K. performed animal work and assisted with ELISA experiments. Z.R.W. performed confocal microscopy. A.H.P., A.K., A.O., R.S.V., C.L.D., J.D. and G.D.L. designed and performed the microneutralisation assay. R.D.L.N. and D.F.C. designed the anti-ZIKV envelope R34 antiserum. A.H.P., A.K., L.D., G.R.S., D.H.B., S.C.G. and A.S.H. provided vital characterised reagents and conceptual support. All authors read and commented the manuscript. C.L.-C. and A.R.S. wrote the manuscript.

## Additional information

**Competing interests:** A.R.-S. and C.L.-C. are co-inventors of the Zika vaccines described in this manuscript, filed by Oxford University Innovation Limited in the International Patent Application No. PCT/GB2017/052220 Zika Vaccine; A.V.S.H. and S.C.G. are co-inventors on a patent application (WO/2012/172277) on the ChAdOx1 viral vector filed by Oxford University Innovation. The remaining authors declare no competing interests.

