## [Peer Review File · Nature Communications]

Reviewers' comments:

Reviewer #1 (Zika)(Remarks to the Author):

Lopez-Camacho and colleagues report the efficacy results of chimpanzee adenovirus vector-based Zika vaccine. They showed prevention of Zika virus infection in mice that were pre-immunized with the vaccine. Although the topic is interesting and important, this reviewer is concerned about several important gaps that are missing in the current manuscript.

1. The authors should provide traditional PRNT values for the immunized mice from the immunized BALB/C mice. This is critical to evaluate the correlation between efficacy and functional neutralization titers.

2. The authors should provide a cohesive data set from the same experimental system. In the current manuscript, the authors performed efficacy experiments in BALB/c, but provided the non-traditional neutralization titers of sera collected from immunized CD1 mice. These are key experiment results and should not be disjointed by piecing the results from two types of mice.

3. The "new" neutralization assay is not clearly described.

4. The authors are over "marketing" the study. This is a mouse efficacy study and the presented results do not conclude "Phase I clinical trial is scheduled to start in 2017." The next logical step is to test its efficacy in NHP model.

5. It is interesting that no ADE was observed in the U937 cell-based in vitro assay. However, the author should temper the relevance of the cell culture results.

Reviewer #2 (Virus, RNA vaccine)(Remarks to the Author):

Numerous Zika vaccines have recently been reported demonstrating protective efficacy in mice and/or non-human primates. Most of these vaccines elicited neutralizing antibody responses after one or more immunization(s) without any adverse events. As Zika virus and dengue virus often co-circulate, it is critical to test preclinical Zika (and dengue) vaccines for antibody-dependent enhancement (ADE) that can induce dengue hemorrhagic fever, a serious disease with poor outcome and potentially worsen the disease course or manifestations of Zika. Lopez-Camacho et al designed 5 DNA and 6 chimpanzee adenoviral (ChAdOx1) vectored Zika vaccines using various portions of the Zika virus genome. Of those, several ChAdOx1 vaccines quickly induced long-lasting T and B cell immune responses and one of them provided sterilizing immunity against Zika virus after a single non adjuvanted immunization. Most importantly, (1) ChAdOx1 vaccines elicit antibodies that can bind to dengue virus envelope raising the possibility that the present vaccine can protect from both Zika and dengue virus infections, and (2) none of the presented vaccines induced in vitro ADE to DENV2 that highly increases the safety profile of those vaccines.

Main criticisms:

There are major problems with the experimental design that need to be addressed:

- The authors do not show NAb responses in Balb/c mice or E-specific IgG in CD1 mice, which creates a problem for the interpretation of the data given that Ext Data Figure 3C shows marked differences in the antibody responses in these two mouse strains. Both analyses should be performed in both mouse strains.
- The authors offer a new and interesting take on the neutralization assay, which may help in making this analysis more high-throughout. However, they do not compare it to any of the more standard assays conducted in this field, such as a traditional plaque reduction neutralization test, which is necessary to compare this vaccine with any of the other promising Zika vaccine candidates. It is known that very large differences in titer is observed between different neutralization assays. A comparison of their new assay to standard and published assays would greatly improve the paper.
- Many of the figures and assays are not described in enough detail for the reader to understand

what was done, and the discussion could be much improved, as outlined below.

Other important concerns:

- Figure 1A: Something seems wrong with this tree - there should be much greater distance between "Other Flaviviruses" and Zika than between different Zika strains. We also can't read the text.
- Figure 1B: There are no serum dilutions stated for these ELISAs, so this data cannot be interpreted.
- Figure 3B: A linear, three-segment y-axis on these graphs is very hard to read. A log scale would be much clearer.
- Ext Data Figure 3C: No dilutions given for ELISA.
- Figure 3E: Please provide a clearer definition for "viremia peak ablation" in either the figure legend or Methods.
- Figure 4A: If I am reading this correctly, you have averaged together the neutralization curves for all mice in each group. Better would be to show the individual curves for each mouse (with error bars if possible), and/or to plot the MN50 value for each mouse, so readers can see the full variation and complexity of the data. This is needed given your statement on line 143 that neutralization was only seen in a few mice in some groups.
- Figure 4B: What is the capture antigen (and its source) in these ELISAs—DENV E protein, whole DENV virus, or a synthetic peptide of residues 147-166?
- Intro/Discussion: The discussion of this report in the context of the Zika vaccine field could be much improved. Please reference the other Zika vaccine papers, including Muthumani et al, *npj Vaccines* 2016; Kim et al, *EBioMedicine* 2016; Pardi et al, *Nature* 2017; and especially Richner et al, *Cell* 2017, which directly addresses the issue of ADE, a major topic of your manuscript. The latter two papers additionally discuss mRNA-based vaccines, a platform which is not mentioned in your intro or discussion. Please discuss how your vaccine compares in neutralizing titer and other characteristics (e.g. species specificity) with these other vaccines, in particular RhAd52.
- Ext Figure 3A: The x-axis here is not labeled clearly with respect to the mouse strains (the middle groups fall under neither label). And are prM and E the antigens used for stimulation? Please also provide the source of the peptide pools used.

Minor concerns:

- Lines 64-65: Grammar error
- Ext Fig 1: Define the root of these trees.
- Fig. 1B: This figure reads OD405 on the y-axis but the methods say OD450. Please correct.
- Lines 108-109 and throughout: "Sterile" protection might not be the most accurate term when serum viremia is undetectable, in the absence of testing other highly infected tissues (such as spleen) or looking at anamnestic antibody responses in response to viral challenge. Better to be more specific, e.g. "protected from detectable viremia"
- Line 129-130: I am not sure what you mean by "complete, intermediate, and low sterile protection." Which adjectives correspond to which of the six vaccine conditions listed?
- Line 336: Do you mean "neutralized 50% of infection" rather than ">50%"?
- Line 138: "signal-to-noise ratio of >10" – It is not clear how this was calculated. This may be better expressed in terms of a standard deviation or as error bars or a plot of all data points in Figure 4A.
- In the challenge experiment, no information about the injected vaccine dose can be found in the text. This should be added to Fig. 3 legends.

Reviewers' comments:

Reviewer #1 (Zika)(Remarks to the Author):

Lopez-Camacho and colleagues report the efficacy results of chimpanzee adenovirus vector-based Zika vaccine. They showed prevention of Zika virus infection in mice that were pre-immunized with the vaccine. Although the topic is interesting and important, this reviewer is concerned about several important gaps that are missing in the current manuscript.

1. The authors should provide traditional PRNT values for the immunized mice from the immunized BALB/C mice. This is critical to evaluate the correlation between efficacy and functional neutralization titers.

We aimed to establish a high-throughput methodology to allow a rapid assessment of ZIKV vaccine efficacy *in vitro*; especially when assessing many vaccine candidates using dilutions and various time points with 3 replicates. Although we acknowledge PRNT is a traditional assay, we have seen that this assay displays lab-to-lab variability, especially when using different ZIKV strains. We hope the reviewer appreciates the vaccine efficacy achieved with our candidates, using the same ZIKV challenge model as Larocca *et al.*, which is now being used in several papers. Moreover, we have reached similar endpoint titers as the ones reached by Larocca *et al.*, which mirrored protection (viral loads) observed in our study.

Although PR is a traditional method, Larocca *et al.* have reported a neutralisation assay in Nature (Nature 536(7617):474-8. 2016), using a Polymerase Chain Reaction-based technique. Our method is highly similar to the methodology reported by Larocca *et al.* (MN50) with a difference consisting of assessing the Envelope ZIKV protein expression, rather than assessing mRNA levels. The technique was used for BALB/c sera and it is reported in our manuscript (Line 132-150 and Figure 4).

2. The authors should provide a cohesive data set from the same experimental system. In the current manuscript, the authors performed efficacy experiments in BALB/c, but provided the non-traditional neutralization titers of sera collected from immunized CD1 mice. These are key experiment results and should not be disjointed by piecing the results from two types of mice.

We agree with this comment and we now provide neutralisation (NT) results both, in BALB/c and CD-1 (Lines 151-161, highlighted). Importantly we have carried out the assay shortly and long after vaccination using two time points of 4 and 16 weeks to assess memory responses. CD1 NT is provided as an extended legend and we believe that the NT assay from outbred mice such as CD1,

which is not limited to specific HLA genes, is biologically meaningful.

3. The “new” neutralization assay is not clearly described.

We describe the methodology and results in lines 132-161 and include a detailed description in Material and Methods in line 365-376 (highlighted).

4. The authors are over “marketing” the study. This is a mouse efficacy study and the presented results do not conclude “Phase I clinical trial is scheduled to start in 2017.” The next logical step is to test its efficacy in NHP model.

We appreciate this concern and we have decided to exclude this statement. Pointing out that our platform is suitable for further pre-clinical and clinical testing. Importantly, our development has received funding from the British Government to be taken to clinical trials in 2018 and the vaccine is currently being produced in GMP. Although assessment in macaques would be desirable, this has not been a requisite to obtain funding for a phase I trial in Oxford but we agree it is a logical step to take and we are considering this and applying for funds to test the most efficacious vaccine reported in this paper.

5. It is interesting that no ADE was observed in the U937 cell-based *in vitro* assay. However, the author should temper the relevance of the cell culture results.

We appreciate this feedback. We rephrased the conclusions and tempered the description of our results (Line 173-177).

Reviewer #2 (Virus, RNA vaccine)(Remarks to the Author):

Numerous Zika vaccines have recently been reported demonstrating protective efficacy in mice and/or non-human primates. Most of these vaccines elicited neutralizing antibody responses after one or more immunization(s) without any adverse events. As Zika virus and dengue virus often co-circulate, it is critical to test preclinical Zika (and dengue) vaccines for antibody-dependent enhancement (ADE) that can induce dengue hemorrhagic fever, a serious disease with poor outcome and potentially worsen the disease course or manifestations of Zika. Lopez-Camacho et al designed 5 DNA and 6 chimpanzee adenoviral (ChAdOx1) vectored Zika vaccines using various portions of the Zika virus genome. Of those, several ChAdOx1 vaccines quickly induced long-lasting T and B cell immune responses and one of them provided sterilizing immunity against Zika virus after a single non adjuvanted immunization. Most importantly, (1) ChAdOx1 vaccines elicit antibodies that can bind to dengue virus envelope raising the possibility that the present vaccine can protect from both Zika and dengue virus infections, and (2) none of the presented vaccines induced *in vitro* ADE to DENV2 that highly increases the safety profile of those vaccines.

Main criticisms:

There are major problems with the experimental design that need to be addressed:

- The authors do not show NAb responses in Balb/c mice or E-specific IgG in CD1 mice, which creates a problem for the interpretation of the data given that Ext Data Figure 3C shows marked differences in the antibody responses in these two mouse strains. Both analyses should be performed in both mouse strains.

We agree with this comment, neutralisation in BALB/c is now provided and importantly, we have extended the analysis to include results in BALB/c on weeks 4 and 16 post immunisation (Lines 132-161). CD1 neutralisation is also provided in the results (Lines 157-161) and as an extended legend (Extended Fig 3B). We believe that the results in outbred mice such as CD1 is biologically meaningful as it is not limited to a few HLA genes,

- The authors offer a new and interesting take on the neutralization assay, which may help in making this analysis more high-throughput. However, they do not compare it to any of the more standard assays conducted in this field, such as a traditional plaque reduction neutralization test, which is necessary to compare this vaccine with any of the other promising Zika vaccine candidates. It is known that very large differences in titer is observed between different neutralization assays. A comparison of their new assay to standard and published assays would greatly improve the paper.

We aimed to establish a high-throughput methodology to allow a rapid assessment of ZIKV vaccine efficacy *in vitro*; especially when assessing many vaccine candidates using dilutions and various time points with 3 replicates. Although we acknowledge PRNT is a traditional assay, we have seen that

this assay displays lab-to-lab variability, especially when using different ZIKV strains. We hope the reviewer appreciates the vaccine efficacy achieved with our candidates, using the same ZIKV challenge model as Larocca *et al.*, which is now being used in several papers. Moreover, we have reached similar endpoint titers as the ones reached by Larocca *et al.*, which mirrored protection (viral loads) observed in our study.

Although PR is a traditional method, Larocca *et al.* have reported a neutralisation assay in Nature (Nature 536(7617):474-8. 2016), using a Polymerase Chain Reaction-based technique. Our method is highly similar to the methodology reported by Larocca *et al.* (MN50) with a difference consisting of assessing the Envelope ZIKV protein expression, rather than assessing mRNA levels. The technique was used for BALB/c sera and it is reported in our manuscript (Line 132-150 and Figure 4).

- Many of the figures and assays are not described in enough detail for the reader to understand what was done, and the discussion could be much improved, as outlined below.

- Figure 1A: Something seems wrong with this tree - there should be much greater distance between "Other Flaviviruses" and Zika than between different Zika strains. We also can't read the text.

We have made the diagrams bigger and also replace it with a simpler version, we have relied on advice from a phylogenetics expert and there seems to be no error, as it can be seen that "other Flaviviruses group" is branched out and separated from ZIKV lineages.

- Figure 1B: There are no serum dilutions stated for these ELISAs, so this data cannot be interpreted.

We are sorry but this is not an ELISA, Figure 1B is a comparison of identity of our vaccine antigen vs Asian lineage sequences. However, we have made sure all the ELISAs presented in the study are clearly explained and described. We have also made sure it is properly described in Material and Methods.

- Figure 3B: A linear, three-segment y-axis on these graphs is very hard to read. A log scale would be much clearer.

We believe that keeping a three-segment y-axis permits a good comparison of the vaccinated groups with the naïve control group. To help with this, we enlarged the figures and make sure these read properly by allowing more space between axis intervals.

- Ext Data Figure 3C: No dilutions given for ELISA.

We have now included more informative graphs in this figure, including serial dilution graphs for the pre-challenge ELISAs.

- Figure 3E: Please provide a clearer definition for "viremia peak ablation" in either the figure legend or Methods.

We now have change it as: "Avoided 3 day viraemia peak", hoping this will be clearer.

- Figure 4A: If I am reading this correctly, you have averaged together the neutralization curves for all mice in each group. Better would be to show the individual curves for each mouse (with error bars if

possible), and/or to plot the MN50 value for each mouse, so readers can see the full variation and complexity of the data. This is needed given your statement on line 143 that neutralization was only seen in a few mice in some groups.

We agree with this, now we have graphed the Microneutralisation (MT₅₀ titre) instead of the dilutions used for the NT assay. This will allow a better appreciation of the variation of each mouse serum sample for every mouse.

- Figure 4B: What is the capture antigen (and its source) in these ELISAs—DENV E protein, whole DENV virus, or a synthetic peptide of residues 147-166?

Information has been added now in Material and Methods (Line 385-392), these are full DENV1-4 Env proteins from Dundee Cell Products as indicated in the text.

- Intro/Discussion: The discussion of this report in the context of the Zika vaccine field could be much improved. Please reference the other Zika vaccine papers, including Muthumani et al, npj Vaccines 2016; Kim et al, EBioMedicine 2016; Pardi et al, Nature 2017; and especially Richner et al, Cell 2017, which directly addresses the issue of ADE, a major topic of your manuscript. The latter two papers additionally discuss mRNA-based vaccines, a platform which is not mentioned in your intro or discussion. Please discuss how your vaccine compares in neutralizing titer and other characteristics (e.g. species specificity) with these other vaccines, in particular RhAd52.

We agree and have now referenced these papers. Various of these were published while our manuscript was submitted and during a 4-month review by Nat Comm. We have now included all this papers on the discussion. We appreciate the feedback.

Minor concerns:

- Lines 64-65: Grammar error

Fixed

- Ext Fig 1: Define the root of these trees.

Fixed

- Fig. 1B: This figure reads OD405 on the y-axis but the methods say OD450. Please correct.

Fixed

- Lines 108-109 and throughout: “Sterile” protection might not be the most accurate term when serum viremia is undetectable, in the absence of testing other highly infected tissues (such as spleen) or looking at anamnestic antibody responses in response to viral challenge. Better to be more specific, e.g. “protected from detectable viremia”

Fixed, we have added this important comment throughout the manuscript. We appreciate the feedback.

- Line 129-130: I am not sure what you mean by “complete, intermediate, and low sterile protection.” Which adjectives correspond to which of the six vaccine conditions listed?

Fixed, we have deleted this and changed to Parameters of protection after challenge.

- Line 138: “signal-to-noise ratio of >10 ” – It is not clear how this was calculated. This may be better expressed in terms of a standard deviation or as error bars or a plot of all data points in Figure 4A.

Fixed, we have used means and error bars.

- In the challenge experiment, no information about the injected vaccine dose can be found in the text. This should be added to Fig. 3 legends.

This info is shown in material and methods and included in the legend.

Reviewers' comments:

Reviewer #1 (Remarks to the Author):

This is a revised manuscript. Although the authors have made some efforts to improve the study, they have not adequately addressed this reviewer's previous concerns.

1. Traditional neutralizing assay was not performed, as requested by both reviewers. These are essential data to evaluate the vaccine and to compare with other vaccine candidates in pipeline.
2. The assay described in lines 142-148 is not clear to this reviewer.
3. For construct Env Δ TM, the sentence from lines 126-129 is not clearly supported by the data shown in Figure 3F.

Reviewer #2 (Remarks to the Author):

The authors addressed most of the previous concerns. The following minor issues were not adequately addressed.

Minor issues:

The authors should provide statistics for Figure 4B and Extended Data Figure 3. The best would be to add a section about Statistical Analyses to the Methods, too.

There are some typos on Figures 3F and 4A. These should be fixed.

Reviewer #3 (Remarks to the Author):

This revised manuscript by Lopez-Camacho and colleagues has addressed most of the concerns raised by reviewers on the original submission. The manuscript now shows that a recombinant chimpanzee adenoviral vector vaccine for Zika virus (ZIKV) is both immunogenic and protective against wild-type ZIKV challenge in BALB/c mice. Specifically, they also show that deletion of the transmembrane region of the prME construct produced better vaccination outcome than full prME gene constructs. The authors suggest that the ChAdOx1 ZIKV vaccine without the transmembrane anchors could be further explored in preclinical and clinical trials.

A couple of areas require additional attention:

1. The explanation on why ChAdOx1 prME and E constructs did not perform as well as ChAdOx1 prME Δ TM and E Δ TM, respectively, is incomplete. The authors showed that the cellular localization of the ZIKV structural proteins are different, depending on the presence or absence of the transmembrane region of the proteins. However, there is no data to show that these differences in cellular localization resulted in differences in antigen secretion or HLA presentation and thus contribute to the observed differences in immunogenicity and vaccine efficacy. Measuring antigen abundance in the culture supernatant of transfected cells would be an informative data to include in this paper.

2. The suggestion that the vaccine-induced antibodies do not enhance DENV infection is problematic. The authors reacted mouse antibodies with ZIKV and then inoculated this mixture onto human U937 cells. It is well established now that there is some degree of Fc-Fc receptor incompatibility between mouse Fc and human Fc receptors that reduces the degree of ADE, in vitro. What is also missing in this experiment is a group of control mice inoculated with a heterologous serotype of DENV before challenge with DENV-2. The evidence to support the notion that vaccination of ChAdOx1 would not produce in DENV enhancing antibodies is thus weak. The

authors should include this caveat in their discussion and moderate the strength of their claim on ADE.

3. The use of BALB/c mice for ZIKV infection studies is somewhat limited by the lack of disease outcome in these mice. Critically, any vaccine against ZIKV must be able to prevent infection in key organs where the virus can either persist, such as the testes, or more rarely result in life-threatening disease, such as the CNS. Most groups investigating Zika have used the A129 mice, which is deficient in type-I interferon receptor. Although how well prevention of disease or persistent ZIKV infection in A129 extrapolates to similar protection in humans is unclear, demonstration of the effectiveness of ChAdOx1 in preventing such outcomes would likely still be necessary for subsequent clinical translation. A discussion on this limitation in data should be considered for this manuscript.

Reviewers' comments:

Reviewer #1 (Remarks to the Author):

1. Traditional neutralizing assay was not performed, as requested by both reviewers. These are essential data to evaluate the vaccine and to compare with other vaccine candidates in pipeline.

We thank the reviewer for the feedback. For this round of review, it is 1 out of 3 reviewers that requested a comparison between MN (microneutralisation) and PRNT (Plaque Reduction Neutralisation titre). All the authors in this manuscript agree that our ZIKV challenge efficacy data plus our MN50 assay are strong evidence to support that our vaccine development is effective, and that performing an experimental comparison between other pipeline vaccines is beyond the scope of this manuscript.

We have done an updated revision of the zika vaccines in the pipeline, as the reviewer has kindly suggested, with special attention to vaccines that have completed Phase I clinical trials in humans. For all the trials cited below, the methodology of choice to assess *in vitro* neutralisation is a microneutralisation assay, rather than a PRNT:

- 1) a Phase 1 trial performed at UPENN-Inovio Pharma,
- 2) a Phase 1 trial completed at Walter Reed Army Institute of Research, Silver Spring, MD, USA,
- 3) a Phase 1 trial done at Saint Louis University, MO, USA, and
- 4) a Phase 1 clinical trial performed at Beth Israel Deaconess Medical Center, Boston, MA., USA.

Furthermore, another 5) Phase 1 trial, performed at National Institute of Allergy and Infectious Diseases, National Institutes of Health, Bethesda, MD, used a neutralisation based on a Zika Reporter Virus Particle Assay (GFP) rather than a PRNT.

Below, the references:

- Tebas P, Roberts CC, Muthumani K, et al. Safety and immunogenicity of an anti-Zika virus DNA vaccine — preliminary report. *N Engl J Med*. DOI: 10.1056/NEJMoa1708120
- Modjarrad K, Lin L, George SL, et al. Preliminary aggregate safety and immunogenicity results from three trials of a purified inactivated Zika virus vaccine candidate: phase 1, randomised, double-blind, placebo-controlled clinical trials. *Lancet* 2017; [http://dx.doi.org/10.1016/S0140-6736\(17\)33106-9](http://dx.doi.org/10.1016/S0140-6736(17)33106-9).
- Gaudinski, M. R. et al. Safety, tolerability, and immunogenicity of two Zika virus DNA vaccine candidates in healthy adults: randomised, open-label, phase 1 clinical trials. *Lancet* [https://doi.org/10.1016/S0140-6736\(17\)33105-7](https://doi.org/10.1016/S0140-6736(17)33105-7) (2017).

We appreciate that PRNT is the traditional method to assess ZIKV infectivity *in vitro*. However, we would like to highlight the fact that the ZIKV microneutralisation is not a new methodology and we have done a minor modification to the ELISA-based detection, which actually is more specific and accurate (a sandwich ELISA). Microneutralisation is a standardised assay that has been used in several publications for both, mice and non-human primates' efficacy studies for Zika and other flaviviruses. The following are examples for zika vaccines:

- Abbink, P. et al. Protective efficacy of multiple vaccine platforms against zika virus challenge in rhesus monkeys. *Science* 353, 1129–1132 (2016).
- Yi G. et al. A DNA Vaccine Protects Human Immune Cells against Zika Virus Infection in Humanized Mice *EBioMedicine* 2017 Nov;25:87-94. doi: 10.1016/j.ebiom.2017.10.006. Epub 2017 Oct 6.
- Abbink P, et al. Durability and correlates of vaccine protection against Zika virus in rhesus monkeys. *Science Transl Med* 2017 Dec 13;9(420). pii: eaao4163. doi: 10.1126/scitranslmed.aao4163.
- Xu K. et al. Recombinant Chimpanzee Adenovirus Vaccine AdC7-M/E Protects against Zika Virus Infection and Testis Damage. *J Virol* 2018 Feb 26;92(6). pii: e01722-17. doi: 10.1128/JVI.01722-17. Print 2018 Mar 15.

Due to the limited availability of serum samples, budget and time constraints, and taking into consideration the reviewer suggestion, we have performed a fair head-to-head comparison between our MN assay vs a traditional PRNT assay, using a series of 2-fold dilutions of a highly potent monoclonal neutralising antibody to ZIKV, EDE1 C8 (Characterised in Gavin Sreaton's Lab). Our results demonstrate that our MN assay is more stringent at detecting ZIKV infection than PRNT (see graph below, now included in Extended Figure 2). Therefore, the MN50 values

in our study are moderate when compared to PRNT. Interestingly, the data gathered from this experiment, mimics exactly the NT comparisons from Dowd *et al.* in her Science paper (2016), in which the MN50 titers are actually lower than those of that FRNT (Focus Reduction Neutralisation Titre) 50 assay, which is basically a PRNT50 (see table below).

Group	NHP ID	RVP50		MN50	FRNT50	
		AVE (N=2-4)	STDEV	N=1	AVE (N=1-4)	STDEV
VRC5288 4mg x2	A13V066	6881	783	673	1937	1302
	A13V071	1972	104	102	636	91
	05C043	2600	465	953	2175	1985
	A13V042	2788	832	245	529	--
	22612	1428	204	163	288	17
	14012	5816	2443	1597	2908	2196

*Table extracted from Supplementary Results (Table S1) Dowd et al, Science 2016.

Despite PRNT50 is a traditional method, there are numerous vaccine developments that are not using PRNT but VLP-ZIKV; and using many types of reporters, such as luciferase, mCherry or GFP. Some of them used flow cytometry and others have used immunofluorescence to assess infected cells. These methodologies, in addition to Focal Reduction NT and PRNT show great variability between published papers as they used different ZIKV strains, different ZIKV purification methodologies and different cell lines (i.e Vero, BHK-21). Therefore, it will be challenging to reach a meaningful comparison to all vaccines in the pipeline but to compare vaccine developments head to head, which is even more challenging.

We have included most of these considerations as a new paragraph in the discussion section.

2. The assay described in lines 142-148 is not clear to this reviewer.

Thank you very much for highlighting this issue, we have now corrected it and made it much clearer. We also have updated the Material and Methods accordingly.

3. For construct Env Δ TM, the sentence from lines 126-129 is not clearly supported by the data shown in Figure 3F.

Thank you for this, we have rephrased now the sentence. In an effort to connect the subcellular localisation with antigen expression, we are now including a thorough analysis of expression in both intra- and extracellular compartments for all the immunogens used in this study, which in turn can support antigen secretion and availability after vaccination. The following diagram has been added to Fig 3G and will complement the immunofluorescence data Fig 3F.

Also, we have assessed the retention within the cells of the antigens in early expression times, which now has been included as Extended Figure 6 (see below). We are confident that addition of these data will positively improve our manuscript.

Assessment of early ZIKV expression in cells

40 hour transfection

Reviewer #2 (Remarks to the Author):

The authors addressed most of the previous concerns. The following minor issues were not adequately addressed.

Minor issues:

The authors should provide statistics for Figure 4B and Extended Data Figure 3. The best would be to add a section about Statistical Analyses to the Methods, too.

We thank the reviewer for the feedback. We have included a One-Way ANOVA, with multiple comparisons to Figure 4 and Figure 3.

For the case of Figure 4B, assessing significance between groups is not ideal since we are using 5 different antigens to coat our ELISA plates, we aimed to provide a qualitative assessment into which extent anti-ZIKV antibodies are able to cross-react with dengue serotypes.

There are some typos on Figures 3F and 4A. These should be fixed.

Thank you, we have gone through Fig 3 and 4 to check for typos. As well we have a member of our staff that will assist to ensure that there are no grammatical errors, after the paper has been accepted.

Reviewer #3 (Remarks to the Author):

This revised manuscript by Lopez-Camacho and colleagues has addressed most of the concerns raised by reviewers on the original submission. The manuscript now shows that a recombinant chimpanzee adenoviral vector vaccine for Zika virus (ZIKV) is both immunogenic and protective against wild-type ZIKV challenge in BALB/c mice. Specifically, they also show that deletion of the transmembrane region of the prME construct produced better vaccination outcome than full prME gene constructs. The authors suggest that the ChAdOx1 ZIKV vaccine without the transmembrane anchors could be further explored in preclinical and clinical trials.

A couple of areas require additional attention:

1. The explanation on why ChAdOx1 prME and E constructs did not perform as well as ChAdOx1 prME Δ TM and E Δ TM, respectively, is incomplete. The authors showed that the cellular localization of the ZIKV structural proteins are different, depending on the presence or absence of the transmembrane region of the proteins. However, there is no data to show that these differences in cellular localization resulted in differences in antigen secretion or HLA presentation and thus contribute to the observed differences in immunogenicity and vaccine efficacy. Measuring antigen abundance in the culture supernatant of transfected cells would be an informative data to include in this paper.

Dear Reviewer, thank you very much for raising this important issue. We have now performed a thorough analysis of antigen expression and have included the results in two figures (Figure 3G and Extended Figure 6), and the description of results in the manuscript. Importantly, we have demonstrated that prME Δ TM and Env Δ TM are able to secrete ZIKV Env antigen to the culture supernatant with higher efficiency of their counterparts Env and prME. We are including the following panel to Figure 3, to further support the immunofluorescence data, and although it is shown above, we considered important to reply to each reviewer:

Hours after transfection

In addition, the following panels (see below) are included as an Extended Figure 6, further demonstrating the ability of our antigens to express and secrete. Taken together, we can strongly correlate the efficacy and immunogenicity of our vaccines by modulating ZIKV Envelope membrane anchoring. We thank the reviewer for this feedback, it was indeed really useful to improve our manuscript.

Assessment of early ZIKV expression in cells

24 hour transfection

40 hour transfection

2. The suggestion that the vaccine-induced antibodies do not enhance DENV infection is problematic. The authors reacted mouse antibodies with ZIKV and then inoculated this mixture onto human U937 cells. It is well established now that there is some degree of Fc-Fc receptor incompatibility between mouse Fc and human Fc receptors that reduces the degree of ADE, in vitro.

What is also missing in this experiment is a group of control mice inoculated with a heterologous serotype of DENV before challenge with DENV-2. The evidence to support the notion that vaccination of ChAdOx1 would not produce in DENV enhancing antibodies is thus weak. The authors should include this caveat in their discussion and moderate the strength of their claim on ADE.

Thank you for raising this concern. We are using a human cell line with Fc receptor, and we are testing mice sera. We have included Extended figure 5 showing the ADE *in vitro* assay by using an anti-DENV2 mouse monoclonal (4G2) antibody as a positive control, in U937 cells. This control provides evidence that that Fc-Fc receptor incompatibility is not an issue in our assay (see graph below). However, we agree with the reviewer it will be a good idea to try a group control mouse inoculated with a heterologous serotype of DENV before challenge with DENV-2. However, this experiment is exceptionally time-consuming and technically difficult as our lab in Oxford is not equipped to handle Category 3 pathogens. We have therefore included this concern and temper the ADE data in the discussion. Thank for your feedback.

3. The use of BALB/c mice for ZIKV infection studies is somewhat limited by the lack of disease outcome in these mice. Critically, any vaccine against ZIKV must be able to prevent infection in key organs where the virus can either persist, such as the testes, or more rarely result in life-threatening disease, such as the CNS. Most groups investigating Zika have used the A129 mice, which is deficient in type-I interferon receptor. Although how well prevention of disease or persistent ZIKV infection in A129 extrapolates to similar protection in humans is unclear, demonstration of the effectiveness of ChAdOx1 in preventing such outcomes would likely still be necessary for subsequent clinical translation. A discussion on this limitation in data should be considered for this manuscript.

This is a very interesting and important point. We acknowledge that our current model may be limited by the fact that we are looking only at viremia, and it is now added into the discussion, as requested. Thank you very much for the feedback, coincidentally we are working now in collaboration with Public Health England to assess efficacy of our vaccine candidates in a lethal challenge model, using A129 mice. As the reviewer suggest, we will look for viral loads in several organs, including the brain as well as testing a vaccine specific to African lineage. This will be another paper we expect to publish in the near future.

REVIEWERS' COMMENTS:

Reviewer #1 (Remarks to the Author):

The authors have reasonably addressed this reviewer's comments.

Reviewer #2 (Remarks to the Author):

The authors have modified the manuscript to address most but not all of the concerns raised, but it does not appear they will address all of them.

Reviewer #3 (Remarks to the Author):

The authors have provided detailed and satisfactory responses to the questions raised in the previous review. The manuscript now makes a compelling case for translational studies on this vaccine construct for potential clinical development.